# Two-year death prediction models among patients with Chagas Disease using machine learning-based methods

**Ariela Mota Ferreira**[1]\*, **Laércio Ives Santos**[2], **Ester Cerdeira Sabino**[3], **Antonio Luiz Pinho Ribeiro**[4], **Léa Campos de Oliveira-da Silva**[3], **Renata Fiúza Damasceno**[1], **Marcos Flávio Silveira Vasconcelos D'Angelo**[5], **Maria do Carmo Pereira Nunes**[4], **Desirée Sant ´Ana Haikal**[1]

1 Graduate Program in Health Sciences, State University of Montes Claros (Universidade Estadual de Montes Claros), Montes Claros, Minas Gerais, Brazil, 2 Instituto Federal do Norte de Minas Gerais, Montes Claros, Minas Gerais, Brazil, 3 Institute of Tropical medicine, University of São Paulo (Universidade de São Paulo), São Paulo, São Paulo, Brazil, 4 Department of Internal Medicine, Federal University of Minas Gerais (Universidade Federal de Minas Gerais), Belo Horizonte, Minas Gerais, Brazil, 5 Department of Computer Science, State University of Montes Claros (Universidade Estadual de Montes Claros), Montes Claros, Minas Gerais, Brazil

\* arielamota@hotmail.com

**Data Availability Statement:** All relevant data are within the manuscript and its Supporting Information files.

## Abstract

Chagas disease (CD) is recognized by the World Health Organization as one of the thirteen most neglected tropical diseases. More than 80% of people affected by CD will not have access to diagnosis and continued treatment, which partly supports the high morbidity and mortality rate. Machine Learning (ML) can identify patterns in data that can be used to increase our understanding of a specific problem or make predictions about the future. Thus, the aim of this study was to evaluate different models of ML to predict death in two years of patients with CD. ML models were developed using different techniques and configurations. The techniques used were: Random Forests, Adaptive Boosting, Decision Tree, Support Vector Machine, and Artificial Neural Networks. The adopted settings considered only interview variables, only complementary exam variables, and finally, both mixed. Data from a cohort study with CD patients called SaMi-Trop were analyzed. The predictor variables came from the baseline; and the outcome, which was death, came from the first follow-up. All models were evaluated in terms of Sensitivity, Specificity and G-mean. Among the 1694 individuals with CD considered, 134 (7.9%) died within two years of follow-up. Using only the predictor variables from the interview, the different techniques achieved a maximum G-mean of 0.64 in predicting death. Using only the variables from complementary exams, the G-mean was up to 0.77. In this configuration, the protagonism of NT-proBNP was evident, where it was possible to observe that an ML model using only this single variable reached G-mean of 0.76. The configuration that mixed interview variables and complementary exams achieved G-mean of 0.75. ML can be used as a useful tool with the potential to contribute to the management of patients with CD, by identifying patients with the highest probability of death.

**Funding:** The SaMi-Trop cohort study is supported by the National Institute of Health (NIH), (P50AI098461-02 and U19AI098461-06). ECS received the source of funding for the work. The funders had no role in study design, data collection and analysis, decision to publish, or preparation of the manuscript.

**Competing interests:** The authors have declared that no competing interests exist.

**Trial Registration:** This trial is registered with ClinicalTrials.gov, Trial ID: NCT02646943.

## Author summary

Chagas disease (CD) is a public health problem despite the partial control of its transmission. Up to 30% of infected people may have cardiac alterations, which are associated with a worse prognosis, with high mortality rates. One of the strategies that can be used to define interventions in order to reduce the impact of CD would be Machine Learning (ML). Thus, the aim of this study was to evaluate different models of ML to predict death in two years of patients with CD. We included 1,694 patients with CD, considering 21 municipalities in endemic regions in Brazil over a two-year period. Of these, 7.9% died. Our study revealed that it is possible to develop ML models which allows the development of tools to predict death within two years, among patients with CD. The different techniques ranged G-mean from 0.59 to 0.77. Thus, we observed that ML can be used as a useful tool with the potential to contribute to the management of patients with CD worldwide, by identifying patients with a higher probability of death.

## Introduction

Chagas disease (CD) is recognized by the World Health Organization as one of the thirteen most neglected tropical diseases in the world, and is caused by the protozoan *Trypanosoma cruzi* (*T.cruzi*) and remains a public health problem despite the partial control of its transmission [1–3]. Most patients with CD remain in the "chronic indeterminate form", defined as persistent asymptomatic infection without cardiac or gastrointestinal tract involvement [4]. However, up to 30% of chronically infected people may have cardiac abnormalities, which is the most serious complication of CD [5]. This condition is associated with a worse prognosis, with higher mortality rates compared to other causes of heart failure [2,6].

CD is the leading cause of disability-adjusted life years (DALY) lost among all neglected tropical diseases [7]. Previous studies estimate that more than 80% of people affected by CD in the world will not have access to diagnosis and continued treatment, which partly supports the high morbidity and mortality rate and the social cost of the disease [4,8].

One of the strategies that can be used to define interventions in order to reduce the impact of CD would be Machine Learning (ML). ML is a field of Artificial Intelligence based on computational algorithms that allow computers to learn directly from data, without being explicitly programmed [9]. ML algorithms analyze large volumes of data represented by many characteristics (predictor variables) in reasonable time, and can handle complex relationships between data, which makes them as accurate or more accurate than human specialists in some situations [10]. Thus, ML can be defined as a set of tools and methods to identify patterns in data. These patterns can be used to increase our understanding of a specific problem, make predictions about the future and contribute to decision making. We say that the algorithm learns from the data. Using different configurations, the goal is to find a model that better explains the dataset [11]. However, ML systems, commonly complex, need to be explainable so that the solutions suggested by the models can be understood. [12].

The use of ML to predict deaths from specific diseases has been of increasing interest in the scientific literature [12], however, only one study was located using ML to predict death from

CD. The study investigated whether heart variability could predict death using a sample of 150 CD patients. The model obtained, which used electrocardiogram (ECG) variables, achieved a death prediction power of 95% [13]. However, information from complementary exams, such as the ECG, is not easily accessible in many remote regions where CD is endemic. And no previous studies were identified that used ML adopting easily accessible predictors, such as information from the interview.

The aim of this study was to evaluate different models of ML for predicting death in two years of patients with CD. Models were tested with three different configurations: only with interview variables, only with variables from complementary exams, and finally, with interview variables and variables from complementary exams.

## Method

### Ethical approval

Ethical approval was granted by ethics committees (Research Ethics Committee of the Faculty of Medicine of the University of São Paulo—protocol number: 042/2012; Research Ethics Committee of the State University of Montes Claros—protocol number: 2,393,610; and the National Institutional Review Council (CONEP), number: 179,685/2012). All subjects agreed to participate in the study and signed an informed consent form before starting the collection.

### Studied population

SaMi-Trop is a multicenter study (Trial registration number: NCT02646943, ClinicalTrials.gov) designed by scientists from four Brazilian public universities in the states of Minas Gerais and São Paulo, that established a cohort of carriers of CD, recruited in 21 municipalities that are endemic for CD in the state of Minas Gerais.

In this cohort, patients were selected to participate in the study based on the results obtained from electrocardiogram (ECG) exams, and only patients aged 18 years or older and who had a cardiac abnormality compatible with CD were considered eligible; 4,689 patients were identified, the baseline took place in 2013 and 2014 and included 2,161 participants. Subsequently the first follow-up was in the years 2015 to 2016 with the permanence of 1,709 participants and it was possible to identify 146 deaths, totaling data from 1855 individuals. Details on the recruitment and methodologies of this cohort can be accessed in a previous study [14].

Of the 1855 individuals at follow-up, those with negative or indeterminate serology for CD were excluded, thus all participants included in this study had confirmed serology.

### Data collection

Baseline and follow-up visits were carried out in the public health units of Primary Health Care, where participants were interviewed, had blood samples taken, and ECG tests performed. The interview addressed sociodemographic issues, lifestyle habits, clinical history, CD treatment, physical activity and quality of life. In the follow-up, among other questions, the reason for the patient's loss of follow-up was also addressed, where the alternatives were: death, giving up, and not being located. In all patients, the presence of anti-*T. cruzi* was tested, using microparticle chemiluminescent immunoassay. The negative results were reassessed and the immune-negativity of the result was confirmed by two other chemiluminescence immunoassay tests using different antigens.

The present investigation was conducted with data from the SaMi-Trop cohort, with the predictor variables coming from the baseline, and the outcome coming from the first follow-up.

## Outcome

The outcome "Death" was adopted for this study (no *vs* yes). Usually deaths associated with CD are due to cardiovascular causes, mainly sudden death or secondary to heart failure. In the present study, only 4 non-cardiovascular deaths occurred (one accidental death, two due to cancer and one non-specified death). However, as we were unable to assess the cause of death of each patient, all cause mortality was defined as the endpoint.

## Inclusion and configuration criteria of the models

In this study, the prediction of the outcome "Death" was performed with three different configurations: only with interview variables, only with complementary exam variables, and finally, with interview variables and complementary exam variables simultaneously. In each configuration additional exclusions occurred due to loss of information (missing). It was decided to exclude participants with some missing data and not perform the imputation of these data, because in no configuration did the number of participants with missing data exceed 10% of the total.

## Data pre-processing

The predictor variables of interviews and complementary exams initially considered in the analyses, as well as the way in which they were collected and worked on, are presented in the supplementary material (S1 Table). Among the predictor variables from the interview, 33 variables were considered and among the predictor variables from complementary exams, 15 variables were considered, including information from the ECG, and the test to assess heart failure (NT-proBNP) and the viral load of CD (quantitative PCR).

To choose the format of the NT-proBNP variable, three measures were tested: numerical, categorized by age [15], and categorized with a cutoff point of $\geq 300$ pg/dl [15], the latter being the format that had the best predictive power for the outcome.

## Selection of predictor variables for the model

The selection of variables aimed to reduce the amount of variables initially available, preserving the relevant and discarding the redundant. There are important reasons why variable selection is essential. The procedure reduces the amount of input variables, optimizing professional work time, reducing model training time and overfitting, resulting in a better generalization capability of the models. In this study, we initially selected 5, 10, or 15 most important predictor variables according to the Random Forests ranking, a strategy based on a previous study [16]. In all experiments, the configuration with 10 variables obtained better results and was adopted in this study. However, other simulations using predictor variables of specific interest were also presented (NT-proBNP related).

## Machine learning approaches

Five supervised ML techniques were used separately to predict the death of the cohort participants. The techniques used were: Random Forests [17], Adaptive Boosting [18], Decision Tree [19], Support Vector Machine [20], and Artificial Neural Networks [21]. For training and evaluation of these models, Matlab software version R2015b was used, with 5-fold cross-validation, of which three were used for training (training set), one to adjust the hyperparameters of each technique (validation set) and one to evaluate the performance of the models (test set) (Fig 1). Cross-validation was performed in a stratified manner, i.e., in each of the 5 folders the prevalence of the outcome, in relation to the total number of participants, was preserved.

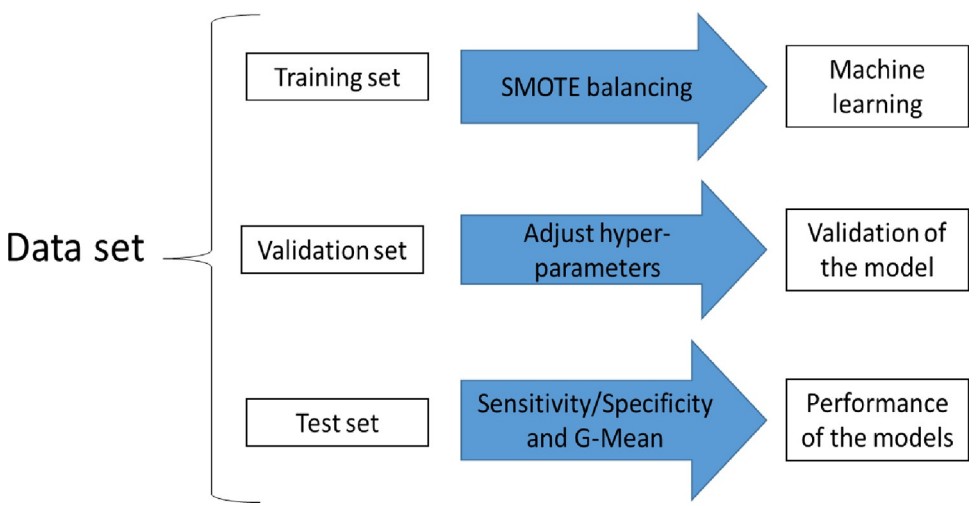

**Fig 1. Flowchart of the process of selecting predictor variables based on cross-validation analysis for predicting death among patients with CD within two years of follow-up.**

Table 1 presents a brief description and the adjusted hyperparameters for each ML technique.

## Class imbalance problem

In all the experiments performed in this study, there was an imbalance in the outcome classes. The number of participants with a positive outcome (death category) was approximately 8%, while the number with a negative outcome (non-death category) was approximately 92%. Thus, it was necessary to insert a balancing step for the training set. The Synthetic Minority Oversampling Technique (SMOTE) method with k = 10 was used to balance the number of instances of the two classes. SMOTE performs a reduced sampling of the majority class and synthesizes new data points in the minority class [22].

## Model performance evaluation

All models were evaluated by different metrics: Sensitivity, Specificity and G-mean. In the context of this study, sensitivity measures the probability of the system predicting a death, given

**Table 1. Description of Machine Learning (ML) techniques adopted in the study.**

| ML Technique | Description | Adjusted hyperparameters |
|---|---|---|
| Random Forests | Uses an aggregate of decision trees from randomly selected instances and predictors. Each tree predicts a problem class and the model prediction is determined by majority vote. | Number of trees, number of sampled predictors, fraction of sampled data. |
| Adaptative Boosting | Uses an aggregate of decision trees from randomly selected instances and predictors in the first tree. From the second tree onwards, the selection of instances is made considering a probability proportional to the prediction error of the previous trees. | Number of trees, Learning Rate. |
| Decision tree | Recursively defined structure composed of decision nodes and leaf nodes. A decision node contains a test on some predictor and for each result of that test there is a link to a subtree. A leaf node corresponds to one of the problem classes. | Maximum number of subdivisions. |
| Support Vector Machine | Searches for a hyperplane that maximizes the distance between instances of two different classes. When the problem dealt with has only two predictors, this hyperplane is represented by a line, and with $n$ predictors a hyperplane with $n$ dimensions is needed to adapt to the data. | Kernel function. |
| Artificial Neural Networks | Simulates the way a human brain learns through artificial neurons. An artificial neuron takes input information from an external source and combines such inputs with non-linear operations producing a result based on the assimilated knowledge. | Number of neurons in the hidden layer, number of epochs and learning rate. |

that the death actually occurred. Specificity measures the probability of the system predicting a non-death, given that the non-death actually occurred. The G-mean equates sensitivity and specificity ($G - mean = \sqrt{(sensitivity * specificity)}$), measuring the balance between the classification performances in the majority and minority classes in problems with class imbalance.

## Statistical analysis

To verify if there are significant differences between the configurations, we performed the Mann Whitney U statistical test, compared the predictive results of each metric (sensitivity, specificity, and G-mean) of the models and considered statistically significant when the p-value < 0.05. Comparison groups were defined as follows: interview variables vs. complementary exam variables; complementary exam variables vs. complementary exam variables without NT-proBNP; complementary exam variables vs. NT-proBNP variable exclusively; complementary exam variables vs. interview variables; and complementary exam variables simultaneously. For all groups we used the values of all folders and the 5 ML models developed.

## Results

Of the 1855 individuals considered in the follow-up of the SaMi-Trop cohort, 161 were excluded because they were negative or indeterminate serology for CD. Therefore, this study considered 1,694 individuals, of which 134 (7.9%) died within two years of follow-up. Fig 2 shows the number of patients included and excluded from the cohort and of eligible patients in the study and in the models obtained.

Table 2 presents categorical predictor variables selected among the 10 with the greatest predictive power in one or more configurations adopted in this study (final model), and their association with death. Quantitative PCR and heart rate variability (numerical variables) were also selected among the 10 variables with the greatest predictive power. The mean PCR in the "non-death" group was 555.40 parasites/mL and in the "death" group it was 666.35 parasites/mL ($p = 0.431$). The mean heart rate variability between the "non-death" group was 304.55 ms and in the "death" group it was 475.97 ms ($p = 0.009$).

## Configuration with interview variables

The 10 most important predictors from the interview (according to Random Forests ranking) were: age, literacy, per capita income, climbing stairs, self-reported ECG irregularities, self-reported skin color, self-reported permanent pacemaker, gender, arterial hypertension, and racing heart. The results of the 5 ML techniques are shown in Fig 3A. In general, the different techniques revealed relatively similar and modest values, with a G-mean of a maximum of 0.64.

## Confuguration with variables from complementary exams

The 10 most important predictor variables from complementary exams were: categorized NT-proBNP, QRS complex duration, heart rate variability, isolated right bundle branch block, quantitative PCR, corrected QT interval, low QRS complex voltage, heart rate, pathological Q Wave, and isolated right bundle branch block plus left anterior fascicular block. Among them, it was observed that the predictive power of the NT-proBNP variable was greater (Fig 4).

The results of the 5 ML techniques considering the complementary exam variables are shown in Fig 3B. All techniques presented G-means above 0.74, the maximum value being 0.77. The configuration considering only complementary exam variables presented predictive power superior to that achieved by configuration with the adoption of only interview variables, according to the U test (p-value < 0.001 for the three metrics).

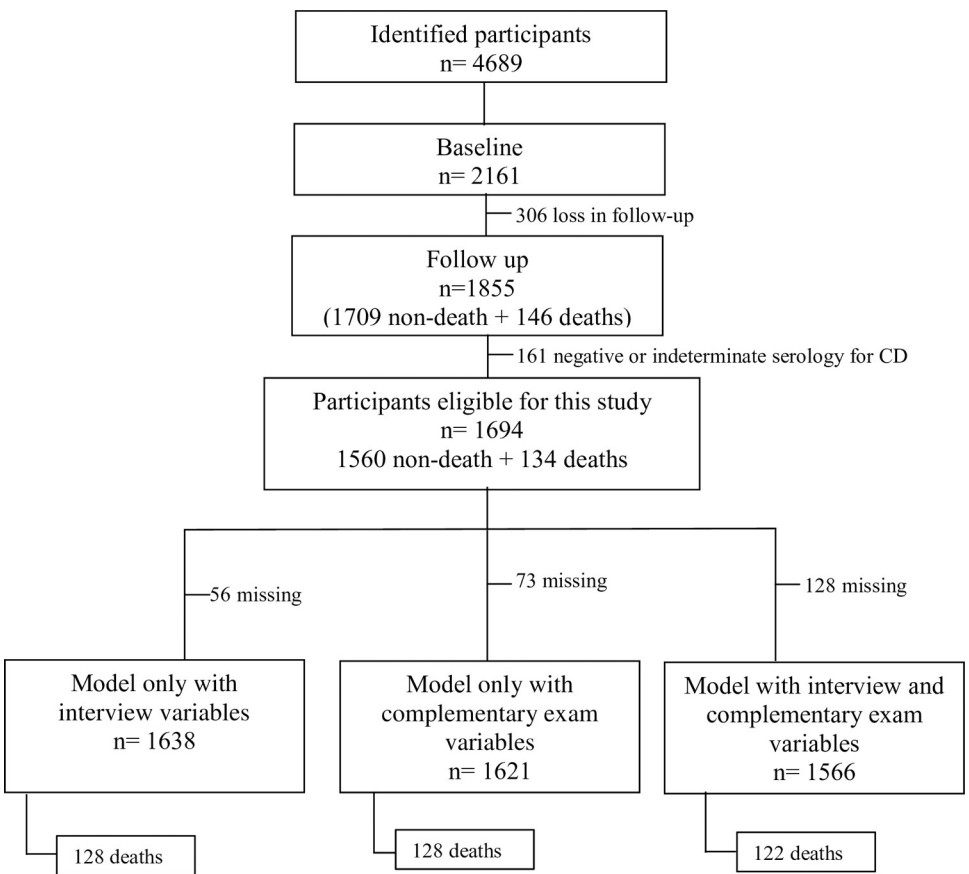

**Fig 2. Flowchart of patients included and excluded from the cohort and of eligible patients in the study and in the models obtained.** SaMi-Trop Project. Minas Gerais.

Due to the high predictive effect of the NT-ProBNP variable, two other configurations were tested: 1) excluding the categorized NT-proBNP variable (only the variables remaining: QRS complex duration, heart rate variability, isolated right bundle branch block, quantitative PCR, corrected QT interval, low QRS complex voltage, heart rate, pathological Q wave, isolated right bundle branch block plus left anterior fascicular block, and pacemaker); 2) Keeping this variable exclusively (Fig 3C and 3D, respectively).

## Configuration with variables from complementary exams excluding the categorized NT-proBNP

Excluding the categorized NT-proBNP variable (Fig 3C), the maximum observed G-mean was 0.66. The reduction observed between this configuration (Fig 3C) and the configuration with complementary exam variables (Fig 3B), according to the U test, shows that there is a significant difference (p-value < 0.002 for the three metrics), revealing a loss in prediction power with the withdrawal of NT-proBNP.

## Configuration with only categorized NT-proBNP

The configuration with NT-proBNP exclusively (Fig 3D), revealed G-mean of up to 0.76 in all techniques, a value similar to the configuration with variables from complementary exams.

**Table 2. Descriptive and bivariate analysis of categorical predictor variables selected among those with the greatest predictive power, and their association with death in patients with Chagas disease (CD).** Minas Gerais, Brazil (n = 1,694).

| Interview variables | Descriptive | Bivariate | | P-value[π] |
|---|---|---|---|---|
| | | Death | | |
| | n (%) | No n (%) | Yes n (%) | |
| **Sociodemographic** | | | | |
| Gender | | | | |
| Male | 562 (33.2) | 503 (89.5) | 59 (10.5) | **0.005** |
| Female | 1132 (66.8) | 1057 (93.4) | 75 (6.6) | |
| Literate* | | | | |
| No | 750 (44.5) | 671 (89.5) | 79 (10.5) | **<0.001** |
| Yes | 937 (55.5) | 884 (94.3) | 53 (5.7) | |
| Age | | | | |
| Up to 60 years | 935 (55.2) | 886 (94.8) | 49 (5.2) | **<0.001** |
| Above 61 years | 759 (44.8) | 674 (88.8) | 85 (11.2) | |
| Self-declared color * | | | | |
| White | 361 (21.4) | 332 (92) | 29 (8.0) | 0.836 |
| Non-white | 1324 (78.6) | 1222 (92.3) | 102 (7.7) | |
| Per capita income* | | | | |
| Greater than R$ 356.33 | 665 (39.7) | 602 (90.5) | 63 (9.5) | **0.033** |
| Less than R$ 356.32 | 1011 (60.3) | 944 (93.4) | 67 (6.6) | |
| **Signs and symptoms reported** | | | | |
| Climb stairs * | | | | |
| No | 617 (36.7) | 538 (87.2) | 79 (12.8) | **<0.001** |
| Yes | 1065 (63.3) | 1010 (94.8) | 55 (5.2) | |
| Self-reported ECG irregularity * | | | | |
| No | 651 (39.2) | 602 (92.5) | 49 (7.5) | 0.559 |
| Yes | 1009 (60.8) | 9258 (91.7) | 84 (8.3) | |
| Racing heart * | | | | |
| No | 603 (36.3) | 602 (92.5) | 49 (7.5) | 0.559 |
| Yes | 1057 (63.7) | 925 (91.7) | 84 (8.3) | |
| **Comorbidities reported** | | | | |
| Arterial hypertension | | | | |
| No | 608 (35.9) | 573 (94.2) | 35 (5.8) | **0.014** |
| Yes | 1086 (64.1) | 987 (90.9) | 99 (9.1) | |
| Permanent self-reported pacemaker * | | | | |
| No | 1559 (93.9) | 1446 (92.8) | 113 (7.2) | **<0.001** |
| Yes | 101 (6.1) | 81 (80.2) | 20 (19.8) | |
| **Variables from complementary exams** | | | | |
| Heart rate* | | | | |
| Normal | 1112 (67.4) | 1024 (92.1) | 88 (7.9) | 0.141 |
| Below normal (up to 59 bpm) | 509 (30.8) | 474 (93.1) | 35 (6.9) | |
| Above normal (above 101 bpm) | 30 (1.8) | 25 (83.3) | 5 (16.7) | |
| Corrected QT interval* | | | | |
| Normal (up to 440 m/s) | 816 (49.4) | 778 (95.3) | 38 (4.7) | **<0.001** |
| Altered (above 441 m/s) | 835 (50.6) | 745 (89.2) | 90 (10.8) | |
| QRS complex duration* | | | | |
| Normal (up to 120) | 959 (58.1) | 911 (95) | 48 (5) | **<0.001** |
| Altered (above 121) | 692 (41.9) | 612 (88.4) | 80 (11.6) | |

*(Continued)*

**Table 2.** (Continued)

| Interview variables | Descriptive | Bivariate | | P-value<sup>π</sup> |
|---|---|---|---|---|
| | | Death | | |
| | n (%) | No n (%) | Yes n (%) | |
| Isolated right bundle branch block plus left anterior fascicular block* | | | | |
| Negative | 1460 (88.4) | 1352 (92.6) | 108 (7.4) | 0.135 |
| Positive | 191 (11.6) | 171 (89.5) | 20 (10.5) | |
| Isolated right bundle branch block* | | | | |
| Negative | 1315 (79.6) | 1209 (91.9) | 106 (8.1) | 0.355 |
| Positive | 336 (20.4) | 314 (93.5) | 22 (6.5) | |
| Pacemaker* | | | | |
| Absent | 1592 (96.4) | 1482 (93.1) | 110 (6.9) | <0.001 |
| Present | 59 (3.6) | 41 (69.5) | 18 (30.5) | |
| Pathological Q waves* | | | | |
| Negative | 1398 (84.7) | 1308 (93.6) | 90 (6.4) | <0.001 |
| Positive | 253 (14.9) | 215 (85) | 38 (15) | |
| Low QRS complex voltage* | | | | |
| Negative | 1556 (94.2) | 1444 (92.8) | 112 (7.2) | 0.001 |
| Positive | 95 (5.8) | 79 (83.2) | 16 (16.8) | |
| Categorized NT-proBNP* | | | | |
| Normal (below 300pg/dl) | 1194 (70.8) | 1166 (97.7) | 28 (2.3) | <0.001 |
| Altered (above 301pg/dl) | 492 (29.2) | 386 (78.5) | 106 (21.5) | |

\* Variation of the n = 1.694 because of missing information.

<sup>π</sup> Chi squared test

There was no significant difference between these settings (p-value > 0.05 for the three metrics).

## Configuration with interview and complementary exam variables

Using interview and complementary exam variables simultaneously, we tested whether there would be an improvement in the prediction of ML models. The results are shown in Fig 3E. Initially, the 20 most important predictor variables were included (10 from interviews and 10 from complementary exams) according to the Random Forests ranking. However, in the final model, 10 variables were kept: categorized NT-proBNP, QRS complex duration, heart rate variability, age, self-reported skin color, stair climbing, isolated right bundle branch block, quantitative PCR, corrected QT interval, and per capita income. In this configuration (Fig 3E), a G-mean of up to 0.75 was observed. In all techniques, no improvement in prediction was observed after using interview variables and complementary exams simultaneously (p-value > 0.05 for the three metrics). Configuration with only complementary exam variables (Fig 3B) had similar predictive power to the configuration with complementary exam and interview variables (Fig 3E).

## Discussion

This study described how ML was able to predict death over a 2-year period in patients with CD, using different techniques and different configurations of predictive models. In general, the prediction of death from CD presented a G-mean value between 0.59 and 0.77, varying according to the techniques and configurations. The five techniques of ML adopted showed

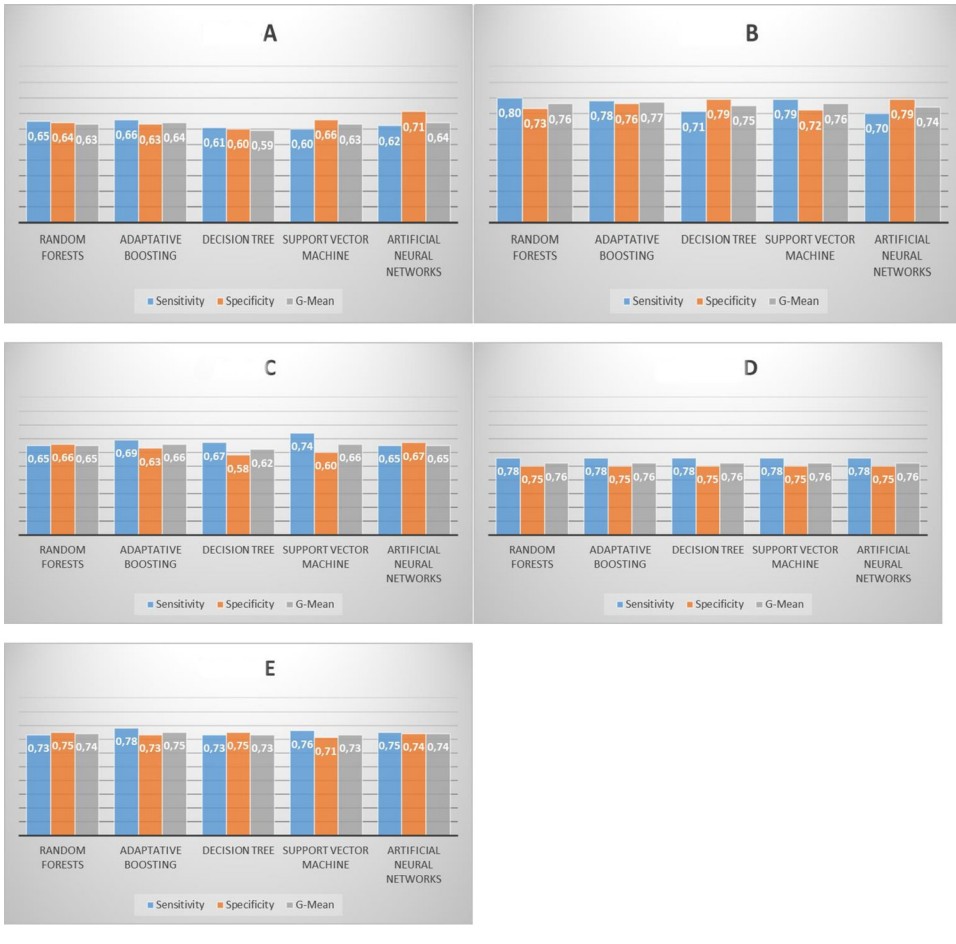

**Fig 3. Performance of models in predicting death for patients with CD, within two years, according to each machine learning technique adopted.** Fig 3A: Considering the interview variables. Fig 3B: Considering the variables of complementary exams. Fig 3C: Considering the variables of complementary exams, excluding the categorized NT-proBNP variable. Fig 3D: Considering the variables of complementary exams, considering only the NT-proBNP variable. Fig 3E: Considering the interview variables and complementary exam variables.

approximately similar values, not being possible to identify superiority of a technique in relation to the others. In the configuration using only variables from the interview, the different techniques revealed relatively modest values, with a maximum G-mean of 0.64. Configuration using only variables from complementary exams revealed a G-mean of up to 0.77. In this configuration, the role of NT-proBNP was evident (Fig 4), showing more than twice the importance observed for the other variables in the model. This finding was confirmed when observing the configuration using only NT-proBNP, where it was observed that this single variable reached a G-mean of 0.76. The configuration of interview variables and complementary exams simultaneously reached a G-mean of up to 0.75. Thus, the three configurations that considered NT-proBNP (variables from complementary exams, only with NT-proBNP, and from interview and complementary exams simultaneously) had similar and superior predictive power to the two configurations (interview and complementary exam variables excluding the NT-proBNP) that did not consider NT-proBNP. The role of this variable in the prediction of death is confirmed by verifying the similarity of the predictive power of the configuration that exclusively adopted this single variable with the predictive power of other more complex configurations.

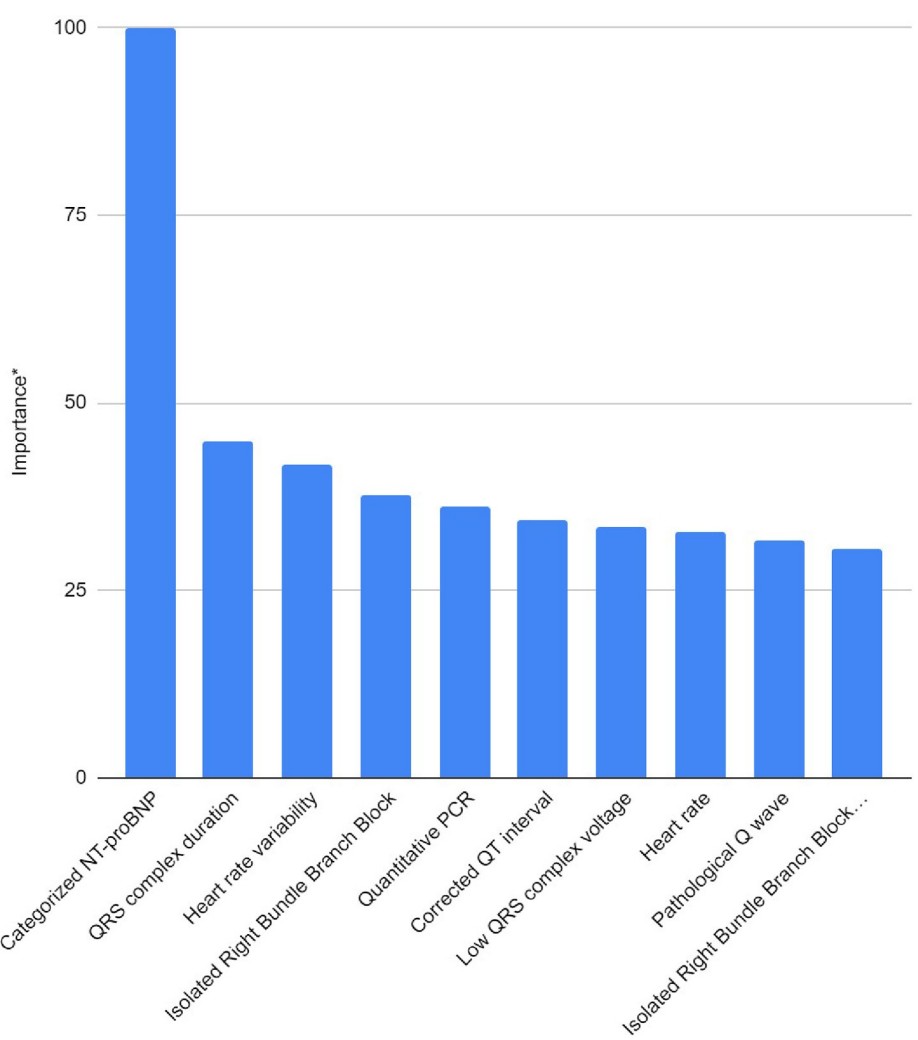

*normalized values on a scale from 0 to 100.

**Fig 4. Importance of predictor variables of complementary exams for predicting death in patients with CD, within two years, according to Random Forests ranking.**

The different models obtained allowed us to identify which configurations have the best predictive power. In addition, the study made it possible to identify, through the use of artificial intelligence, the most important predictors of death from anamnesis, complementary exams, and both.

In the configuration with only variables from the interview, it was noticed that sociodemographic characteristics and self-reported signs/symptoms remained in the model. In this case, the high accuracy of self-reported questions for chronic conditions has already been verified [23]. In terms of clinical projection, adopting only ten interview variables during an anamnesis, individuals who would potentially benefit most from possible previous interventions would be identified. It would be possible to achieve 64% prediction power of death in this configuration. Although the predictive power is considered modest, it can still represent a significant clinical impact. This intervention could be useful in practice, in places where there is restricted access to exams and specialized health services, with a "selective, focused and

exclusive" offering [24]. The literature also reports that the groups that have greater health needs, such as those with CD, are precisely those that have greater difficulty in accessing and using health services [25]. It is a great challenge for the Brazilian public health service (SUS) to achieve equitable access, as each social segment has different demands produced by social processes of exclusion, not always perceived by the government [24]. Thus, the ML tool could contribute to this need for better management of health system users with CD, using simple and effective data, from the anamnesis, especially in remote regions.

In the configuration only with variables from complementary exams, ECG, parasite load (quantitative PCR), and NT-proBNP biomarker variables remained. In this configuration, the prediction power achieved was greater than using only interview variables, highlighting the importance of information of complementary tests for conducting the clinical management of CD. The ECG is the most important complementary exam in the initial evaluation of patients with CD [4]. It is an inexpensive and standardized exam. Detection of *t.cruzi* parasites in blood by PCR has been used to assess the parasite burden and effectiveness of CD treatment, and previous studies have demonstrated the role of PCR in predicting CD progression [26,27]. When verifying the importance of complementary exam variables to compose the model, the NT-proBNP biomarker was prominent. Thus, due to this importance, a configuration with only this variable was tested, and the predictive power of this configuration composed with this single variable was similar to the predictive power of the configuration with the 10 complementary exam variables (76% *vs*. 77%). The configuration with complementary exam variables excluding NT-proBNP had significantly lower predictive power (66%).

NT-proBNP levels already identified are accurate discriminators of heart failure diagnosis, auxiliaries in patient risk stratification, and as powerful predictors of death. A previous study found that the discriminatory ability of NT-proBNP to predict mortality (C = 0.69, 95% CI: 0.66, 0.71) is similar to that of an ECG (C = 0.68, 95% CI: 0.65, 0.71) [28]. Confirming this finding, in another study, in the construction of a risk score to predict 2-year mortality in patients with CD, the NT-proBNP was included, and revealed greater power of death prediction [29]. Furthermore, this measure is the factor most strongly associated with the occurrence of cardiovascular events in the population with CD [30].

Unfortunately, NT-proBNP is not readily available in most clinical settings involving CD. Its large-scale use in the Brazilian public health system as a routine test to aid in management is not envisioned at this time, although it is highly desired, because this exam, performed by a simple blood test, provides fewer echocardiograms (-58.2%) and a reduction in the number of hospitalizations (-12.6%). As it is also a strategy with lower final cost and better diagnostic accuracy, there would be no increase in the budget of the public health system for the diagnosis and treatment of patients with heart failure [31].

Although NT-pro BNP is a quantitative marker, and levels are best interpreted as a continuous variable, cut-off points can still be useful in making its application easy for physicians without extensive experience. Thus, a cut-off of 300 pg/ml is proposed to rule out a diagnosis of heart failure, while higher age-dependent cut-offs are suggested for rule in [15].

Other models for predicting death in the CD population are cited in previous studies. However, only one study was found that used ML to predict death in CD patients. The study included 150 patients and 15 patients who died [13]. Studies that performed this prediction commonly use the methodology of creating risk scores and developed simple models to predict death, with good clinical relevance and a C statistical score from 0.82 to 0.84. All these models shared information from complementary exams to estimate individual risk [29,32,33].

In endemic areas, CD represents a major cause of death from cardiovascular disease [28]. A meta-analysis identified that CD is associated with high mortality, regardless of clinical condition, with a relative risk (RR) of 1.74 (95% CI 1.49–2.03) and attributable risk of 42.5%

considering the exposed group [29]. The region where our study was conducted is characterized by having low demographic density, strong social inequality, large distances between municipalities, and extensive rural areas [34]. In addition to these contextual problems in the region, there is a lack of training, specific knowledge, and safety in the management of patients with CD among primary health care physicians [35,36]. Thus, ML can revolutionize this effective delivery of health care with the advent of new tools and algorithms, developing a new class of smart digital health interventions. However, more studies are needed to demonstrate the effectiveness of digital interventions that rely on machine learning applications in real-life healthcare. More evidence of the clinical utility of ML in the provision of health services is needed. Researchers should go beyond retrospectively validating machine learning models, integrating their models into properly designed digital health tools and evaluating the tools in rigorous studies carried out in real-life environments [37]. Despite their limitations, the use of ML tools could contribute to expanding access to an adequate management of CD, considering that it is an automatic, simple, and inexpensive technique.

Among the strengths of this study, the longitudinal assessment of a large sample of patients with CD who live in endemic areas and in small municipalities, far from the large urban centers commonly portrayed in the studies, stands out. That is, the individuals participating in the investigated sample typically represent populations with CD from endemic areas. But this study is not without limitations. In addition to the difficulties already mentioned in practical implementation, there are some points to comment on, such as the way the models were evaluated in this study, using cross-validation to train and test the models. From a methodological point of view, cross-validation allows for a robust assessment. However, using two independent data sets (one for training and the other for testing the models) would better the generalizability, and application in clinical practice of the models. In addition, independent data are required. A second independent sample must be included for external validation and implementation studies can be carried out to assess the potential impact of using these models to predict death in patients with CD.

## Conclusion

This study evaluated the 2-year predictive power of death in CD patients using different ML settings and techniques. It was possible to develop optimized models, which can contribute with the development of prediction tools. The ML model proved to be useful and with good power to predict death within two years among patients with CD. The different configurations and techniques for predicting death from CD achieved 59% to 77% predictive power. Configuration with information coming only from the interview have the advantage of being used in scenarios with little access to complementary exams, but the incorporation of variables from complementary exams has improved the predictive power. A configuration adopting only the NT-proBNP, in isolation, showed a prediction capacity similar to the best prediction models using interview variables and complementary exams. Thus, the ML method confirmed the role of this biomarker in predicting death. ML can be used as a useful tool with the potential to contribute to the management of patients with CD, by identifying patients with the highest probability of death. However, there is still a need to pursue models with greater predictive power and the clinical implementation of this knowledge through the use of independent data for external validation.

## Supporting information

**S1 TRIPOD Checklist. Prediction Model Development.**
(DOCX)

**S1 Table. Predictor variables initially considered in the analyzes and how they were worked.**
(DOCX)

**S1 Data. Database.**
(XLS)

**S1 Fig.** S1A: Decision tree created from the interview variables and S1B: Decision tree created from the variables of complementary exams.
(TIF)

## Author Contributions

**Conceptualization:** Ariela Mota Ferreira, Laércio Ives Santos, Ester Cerdeira Sabino, Antonio Luiz Pinho Ribeiro, Renata Fiúza Damasceno, Marcos Flávio Silveira Vasconcelos D'Angelo, Maria do Carmo Pereira Nunes, Desirée Sant´Ana Haikal.

**Data curation:** Ariela Mota Ferreira, Laércio Ives Santos, Ester Cerdeira Sabino, Antonio Luiz Pinho Ribeiro, Léa Campos de Oliveira-da Silva, Desirée Sant´Ana Haikal.

**Formal analysis:** Ariela Mota Ferreira, Laércio Ives Santos, Renata Fiúza Damasceno, Marcos Flávio Silveira Vasconcelos D'Angelo, Maria do Carmo Pereira Nunes, Desirée Sant´Ana Haikal.

**Funding acquisition:** Ester Cerdeira Sabino, Antonio Luiz Pinho Ribeiro.

**Investigation:** Ariela Mota Ferreira, Ester Cerdeira Sabino, Antonio Luiz Pinho Ribeiro, Léa Campos de Oliveira-da Silva, Renata Fiúza Damasceno, Maria do Carmo Pereira Nunes, Desirée Sant´Ana Haikal.

**Methodology:** Ariela Mota Ferreira, Laércio Ives Santos, Renata Fiúza Damasceno, Marcos Flávio Silveira Vasconcelos D'Angelo, Maria do Carmo Pereira Nunes, Desirée Sant´Ana Haikal.

**Project administration:** Ariela Mota Ferreira, Ester Cerdeira Sabino, Antonio Luiz Pinho Ribeiro, Léa Campos de Oliveira-da Silva.

**Resources:** Ariela Mota Ferreira, Ester Cerdeira Sabino, Antonio Luiz Pinho Ribeiro, Léa Campos de Oliveira-da Silva.

**Software:** Ariela Mota Ferreira, Laércio Ives Santos, Ester Cerdeira Sabino, Antonio Luiz Pinho Ribeiro, Léa Campos de Oliveira-da Silva.

**Supervision:** Ariela Mota Ferreira, Ester Cerdeira Sabino, Antonio Luiz Pinho Ribeiro, Léa Campos de Oliveira-da Silva.

**Validation:** Ariela Mota Ferreira, Antonio Luiz Pinho Ribeiro, Léa Campos de Oliveira-da Silva, Renata Fiúza Damasceno, Maria do Carmo Pereira Nunes, Desirée Sant´Ana Haikal.

**Visualization:** Ariela Mota Ferreira, Antonio Luiz Pinho Ribeiro, Renata Fiúza Damasceno, Desirée Sant´Ana Haikal.

**Writing – original draft:** Ariela Mota Ferreira, Laércio Ives Santos, Renata Fiúza Damasceno, Marcos Flávio Silveira Vasconcelos D'Angelo, Maria do Carmo Pereira Nunes, Desirée Sant´Ana Haikal.

**Writing – review & editing:** Ariela Mota Ferreira, Laércio Ives Santos, Ester Cerdeira Sabino, Antonio Luiz Pinho Ribeiro, Léa Campos de Oliveira-da Silva, Renata Fiúza Damasceno, Marcos Flávio Silveira Vasconcelos D'Angelo, Maria do Carmo Pereira Nunes, Desirée Sant´Ana Haikal.

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
