## [Decision Letter · Decision Letter 0]

6 Dec 2021

Dear Ferreira,

Thank you very much for submitting your manuscript "Development of a two-year death prediction model among patients with Chagas Disease using methods based on machine learning" for consideration at PLOS Neglected Tropical Diseases. As with all papers reviewed by the journal, your manuscript was reviewed by members of the editorial board and by several independent reviewers. In light of the reviews (below this email), we would like to invite the resubmission of a significantly-revised version that takes into account the reviewers' comments. 

We cannot make any decision about publication until we have seen the revised manuscript and your response to the reviewers' comments. Your revised manuscript is also likely to be sent to reviewers for further evaluation.

Sincerely,

Alberto Novaes Ramos Jr

Associate Editor

Bruce Lee

Deputy Editor

Reviewer's Responses to Questions

**Key Review Criteria Required for Acceptance?**

**Methods**

-Are the objectives of the study clearly articulated with a clear testable hypothesis stated?

-Is the study design appropriate to address the stated objectives?

-Is the population clearly described and appropriate for the hypothesis being tested?

-Is the sample size sufficient to ensure adequate power to address the hypothesis being tested?

-Were correct statistical analysis used to support conclusions?

-Are there concerns about ethical or regulatory requirements being met?

Reviewer #1: -Are the objectives of the study clearly articulated with a clear testable hypothesis stated?

Yes

-Is the study design appropriate to address the stated objectives?

Yes

-Is the population clearly described and appropriate for the hypothesis being tested?

No

-Is the sample size sufficient to ensure adequate power to address the hypothesis being tested?

Yes

-Were correct statistical analysis used to support conclusions?

Yes. Several minor suggestions, two major that will require additional analysis.

-Are there concerns about ethical or regulatory requirements being met?

No concern. Im cool with it. 

The authors cite another paper initially describing the cohort from which the sample was driven. However, it forces the reader to go to this original paper to understand key points such as the inclusion and exclusion criteria and the manner in which these participants were recruited and included. I recommend the authors to clearly state, at least briefly, which were the inclusion and exclusion criteria, which diagnostic tests were performed in Chagas disease diagnosis work-up, and weather the participants were included sequentially in a period or randomly from the original cohort. 

Another issue in the method section is the very short follow-up period for Chagas disease patients. It is clear that this study reports from data of a partial follow-up not yet fulfilled, but I could not find any discussion regarding this short follow-up or expectations about the models’ behaviors with longer follow-up periods in the discussion section. Usually, Chagas Disease patients are followed for 10 or even more years for an outcome to be observed frequently, unless they are in a very advanced stage of (possible heart) disease. Therefore, it is also important to state in the inclusion and exclusion criteria if age or disease severity was considered for this purpose.

The outcome definition is extremely important for appropriate interpretation and applicability of the model. Therefore, I suggest the authors to clearly state if death (yes or no) is an overall CD death, overall heart CD death, any cause death, stroke related death, heart failure death, sudden death or any combination of these, or possible other mechanisms of death. These different mechanisms of death require different prevention and treatment approaches and may drastically change the applicability of the final model. 

“In addition to these, in each configuration additional exclusions occurred due to loss of information (missing). Table 1 shows the number of lost participants, the total number of participants included, in addition to the percentage of death verified in each configuration.” seems result to me. I suggest moving this piece to results section. The method would be how to check missing data, how to decide on data imputation, and what methods are considered to impute data. Additionally, it is not clear what fraction of the excluded participants had the outcome. This could be easily cleared out rearranging table 1 or replacing it by an inclusion and exclusion diagram as recommend in TRIPOD. There is no comment on data imputation. Usually, missing data is not at random and analyzing complete cases only has potential to prediction bias, therefore complete case analysis is considered bad practice. Although there are several complex data imputation algorithms, such as predictive mean matching and ML models, functions to make imputations are currently easy to use in several software, such as STATA and R-project. Of course, if missingness prevalence is very very low, imputation may seem pointless, however there is no way to know that (that Im aware of) until imputation is performed, and complete case data analysis is compared with imputed data analysis, but it does not seem to be the case. As seen in table one, two out of three scenarios have higher prevalence of participants with missing data then participants with the outcome. If missingness is not at random, i.e., it is strongly related to the outcome, then the outcome could be even doubled if missingness had never occurred in first place. How many of the participants with missing data have the outcome data missing?

“To choose the NT-proBNP categorization, three measures were tested: numerical, categorized by age [13], and categorized with a cutoff point of ≥ 300 pg/dl, the latter being the format that had the best predictive power for the outcome.” Although there is reasoning for this procedure this statement is a bit awkward. There is plenty evidence pointing that categorizing continuous predictors is just loss of information and bad practice. By categorizing a continuous predictor in two classes, the authors are reducing the possibility of the machine learning techniques to identify complex data patterns which was mentioned as an advantage in the introduction section. In an extreme example, if BNP is an excellent predictor and may work alone replacing the whole model with a single decision threshold, why workout a machine learning algorithm in the first place? Additionally, similar comments can be worked towards all other predictors shown in table 3 such as PCR and income, and the outcome representation. It is not clear, but it seems that the machine learning techniques are representing the outcome as classes instead of probabilities as it would be expected in regression modeling. This modeling decision may also affect how the predictors may be combined and related to the outcome. Have authors tried regression ML instead of classes ML? Representing the outcome as classes turn the interpretation much easier but it hides the potential prediction error. If the predictions returns probabilities then discussions regarding calibration, decision thresholds and errors around this thresholds would be necessary.

From the supplementary material it is clear that at least 45 predictors were available. Why did the authors chose to perform the models with ten predictors and not 8 or 6 or 12? What is the rationale involved in this decision?

The following suggestion goes beyond the stated research aim. Instead of just comparing different machine learning techniques with binary outcomes the authors could try a comparison of the very same machine learning techniques with binary and survival outcomes. I believe there is a huge knowledge gap on this topic, it would be closer to the already available CD death prediction models in the literature and it would give a more elegant solution to the outcome missing data turning them into censored participants. 

Regarding the model’s performance measures I would suggest the authors to show the performances estimates from both test set and validation set. It is not clear how the subsamples for cross validation were defined. Where they splitted randomly? Or was there a geographic or calendar rationale? Additionally, a brief comment or discussion regarding the performance measures choices or alternatives, such as Youden index, accuracy, total error, positive likelihood ratio in predictive values would be nice.

Reviewer #2: The paper presents the results from the application of a number of supervised machine learning techniques to the problem of estimating the probability of death in patients diagnosed with the Chagas disease. Classical techniques for function approximation like neural networks and support vector machines have been used, together with other standard techniques based on decision-trees. The sample used for the analysis, the SaMi-Trop cohort, consists of ~2000 patients of which ~7% died within two years.

The selection of predictor variables is grouped in three main classes 1) interview; 2) complementary exams; and 3) interview and complementary exams. For model calibration, a 5-fold cross-validation scheme is selected and the performance using the three classes of predictors evaluated. The models are assessed in terms of three standard figures of merit: 1) sensitivity; 2) specificity; and 3) G-mean. The results of the prediction models are discussed and certain conclusions drawn with respect to the importance of the predictors.

Reviewer #3: The authors present a manuscript of relevant interest. The subject falls within the scope of the journal. Overall, the paper is well written and contains valuable information. The bibliography is pertinent and current. However, the text still needs some improvement and minor repairs. Excerpts that deserve special attention in terms of writing were marked in yellow.Comments and additional suggestions have been placed in the margin of the text.

**Results**

-Does the analysis presented match the analysis plan?

-Are the results clearly and completely presented?

-Are the figures (Tables, Images) of sufficient quality for clarity?

Reviewer #1: -Does the analysis presented match the analysis plan?

Yes

-Are the results clearly and completely presented?

Yes

-Are the figures (Tables, Images) of sufficient quality for clarity?

Partially

What is the median follow-up time? What is the range of the follow up time? What is the fraction of participants loss to follow up or with interrupted follow-up? What is the rate of death expressed in events divided by person year (x 100 person-year)? Are participants living mainly in rural or urban areas? These are nice information to allow readers having an idea how the cohort behaves over time and compare with other Chagas disease cohorts. 

Is it possible to show exclusion diagram as a result from the screening and recruiting effort?

Regarding the subsamples for cross validation, it does not make sense estimating the outcome prevalence in all samples if they were splitted randomly, however as it is not clear how does samples were splitted the authors should consider to show the outcome prevalence from all samples.

“In all techniques, no improvement in prediction was observed after using interview variables and complementary exams together. That is, models with only complementary exam variables (Table 04) had similar predictive power to models with complementary exam and interview variables (Table 08).” What is the risk of death for a patient with altered and proBNP? How can a health care provider estimate a risk of death for his/her patient with any of these models?

Reviewer #2: The results are presented and discussed in accordance to the used methodology and some of its limitations have been highlighted. 

A number of qualitative conclusions are made, although little-to-none statistical evidence is provided to support them. Similarly, a number of unclear and subjective statements are made throughout the section regarding the statistical validity, robustness, and ability to generalise of the approach. From this perspective, a moderate use of definitive claims is recommended. The majority of the authors' claims are relevant, but only of qualitative nature and potentially questionable. 

While it is true that it may be possible to use machine learning models in the clinical practice, it is mandatory that the relevance of the outcomes is properly scrutinised and their use in a real decision-making scenario conscientiously disciplined. I believe that strong statements regarding the validity of these methods and their results can only be made if a rigorous protocol is applied along the data collection and analysis pipeline. I am confident that the authors themselves would appreciate a more rigorous practice and degree of conscientiousness in the case they were the patients whose future chances of survival are predicted.

Reviewer #3: Some results are not clearly and completely presented. 

Some tables need minor repairs.

**Conclusions**

-Are the conclusions supported by the data presented?

-Are the limitations of analysis clearly described?

-Do the authors discuss how these data can be helpful to advance our understanding of the topic under study?

-Is public health relevance addressed?

Reviewer #1: -Are the conclusions supported by the data presented?

Partially

-Are the limitations of analysis clearly described?

Yes

-Do the authors discuss how these data can be helpful to advance our understanding of the topic under study?

Partially

-Is public health relevance addressed?

No

“It was possible to develop calibrated models, which allows the development of prediction tools. The ML model proved to be useful…” There is no data supporting this conclusion. Neither in methods section nor in results section there is any mention regarding calibration statistics. Furthermore, it is known that machine learning techniques are usually bad calibrated models and there are several methods proposed to recalibrate the estimated probabilities from these techniques. However, it makes more sense to concern with calibration as a performance dimension with regression models, when probabilities are results of interest. Although the research has produced several models and one final model was considered, the authors did not present a prediction instrument from the final model, therefore stating that the models are useful is not supported by the results as it is. I can’t see how a reader or a health care provider can use any of such models to make predictions. I would expect a formula or a nomogram allowing users to make predictions. However, these are not applicable from machine learning techniques due to the “black box” phenomenon. A work around would be a web calculator. There are many many examples in the Internet of such tools, one I particularly find elegant is found in the following address: https://breast.predict.nhs.uk/ and for chagas disease in the following address: https://shiny.ini.fiocruz.br/pedrobrasil/

Reviewer #2: It is in my opinion difficult to make conclusive statements regarding the appropriateness of the study, as such statements should be based on the statistical properties of the sample and the complexity of the prediction model. 

Because such information is only partially available or missing altogether, I can only assume that the models have been thoroughly optimized and that the class distribution was preserved in the creation of the 5 folds used for validating the hyperparameters. These are the minimal and essential requirements when dealing with statistical information: Their use is rigorously documented in the rich literature that modern public health has generated, it should not be neglected but rather integrated to support the use of machine learning techniques.

My opinion is that accuracies in the 70% range are not outstanding. However, and perhaps more importantly, they highlight the existence of a fundamental structure in the data which should be further discovered and explored. As it is a very uncommon in data analysis that a rigorous statistical treatment is developed. I advise the authors to consider a re-statement of the objectives of their work towards the usual practice to replace explicit statistical information with visual summaries of the data (typically, histograms and scatter plots), and support their presentation and the quality of nevertheless interesting investigations with highly valuable qualitative information. 

I encourage the authors to restructure their contribution with such summaries, as they would introduce the reader to the main features of the used data, rather than their questionable predictive power.

Reviewer #3: The discussion and conclusion related to the parasite load ( quantitative PCR) should be founded better.

The conclusion/discussion about the public health relevance should be reviewed ( see comment on line 353)

**Editorial and Data Presentation Modifications?**

Reviewer #1: (No Response)

Reviewer #2: I do not have any comments about this.

Reviewer #3: The authors present a manuscript of relevant interest. The subject falls within the scope of the journal. Overall, the paper is well written and contains valuable information. The bibliography is pertinent and current. However, the text still needs some improvement and minor repairs. Excerpts that deserve special attention in terms of writing were marked in yellow. Comments and additional suggestions have been placed in the margin of the text. Minor remarks- 

Please see lines 22, 56, 68, 70, 197, 201, 212, 217, 234, 238, 248, 255, 256, 258, and 354.

Major remarks- 

Please see comments on lines 252, 293-294, 321, and 353

**Summary and General Comments**

Reviewer #1: It seems dad the authors where eager to develop prediction models with machine learning techniques. From my point of view, there could be at least two main approaches for this research. The first one would be to extensively test different ways to apply different machine learning techniques using Chagas disease as a case study scenario and discuss particular issues of the modeling process for Chagas disease data behavior with different models. Here, the model development and how to model would be of major interest. The second approach would be to really build a prediction instrument for different clinical scenarios of Chagas disease health care settings, in a way that health care providers could make predictions from these models. Here, the applicability of one or more models would be of major interest. Unfortunately, the authors fell in between those approaches, not fulfilling neither of those. To fulfill the first one, the suggestion would be to compare neural networks with binary outcome with neural networks with survival outcome, in a similar way to compare random forest with binary outcome with random forest with survival outcome etc. To fulfill the second approach the suggestion would be to construct a prediction instrument from the final model or from a couple of chosen models and make it available for users to make predictions.

It seems that no reporting guideline was followed as there are a few key point that I consider crucial for overall understanding that were poorly described. I would personally use the TRIPOD (https://www.equator-network.org/reporting-guidelines/tripod-statement/) as a reporting guideline, but there are others that could be as useful (https://www.ncbi.nlm.nih.gov/pmc/articles/PMC7538196/) 

Abstract: 

Chagas Heart Disease has three main mechanism of death which are: stroke, sudden death and heart failure. It is not clear which mechanism of death it is used as the outcome. If the overall mortality is the outcome, please state so. From the applicability point of view, it is important to know which mechanism of death are involved in the prediction as these mechanisms require different approaches regarding treatment and to courses of action in prevention. Additionally, overall mortality for chagas disease may have a substantial contribution from mortality not related to CD, as these patients live many years or even decades with this condition. Therefore, knowledge regarding this issue is important for appropriate interpretation and applicability of the final model. 

The main objective of the research is to compare five different machine learning techniques, however in the abstract the authors do not state which techniques are being tested and compared. Please, state what are the models being compared. 

In abstract “Using the predictor variables from the 35 interview, the different techniques achieved a maximum accuracy of 62 % in predicting death…”. Is accuracy in this statement the same as overall correct classification or predictions? In the abstract, the authors stated that are three measures of interest to express the model’s performance, and accuracy is not one of them. Please define accuracy or state the model’s performance as one of the three measures of interest.

Introduction:

The authors state in the introduction that machine learning techniques learn from the data and can make predictions. Although I do agree with the authors, this also applies to regression models that are intended to prediction. The main characteristics of machine learning techniques that make it attractive over regression models are their automated procedures and the ability to capture complex patterns in the data in a way that implementing and updating the model in automated computerized systems make it much easier. However, this implementation does not seem to be the case, thus additional issues rise. Most if not all ML techniques suffer (more or less) from the “black box” phenomena. This means that if the data pilot inserts 30 predictors in a potential model, the algorithm shrinks the effect of the variables with no prediction contribution to zero without actually removing the predictor from the model. If the model is implemented in a way that the user needs to fill the 30 predictors values, it becomes less and less attractive to users as the number of predictors rises. Most of ML are just not made for removing the weak or near zero effect predictors, thus reducing the number of predictors is a workload balance between the data pilot effort and the user effort. This is briefly cited at the methods section “Selection of predictor variables for the model”. But the priorities of the authors are not clearly stated around this issue. 

From the performance point of view, there are plenty studies comparing performance of regressions and machine learning techniques with conflicting results. Additionally, recent systematic review points toward an evidence that there is no clear performance advantage off machine learning techniques over regression models when considering studies last susceptible to bias. Please take a look at the paper below.

Christodoulou E, Ma J, Collins GS, Steyerberg EW, Verbakel JY, Van Calster B. A systematic review shows no performance benefit of machine learning over logistic regression for clinical prediction models. Journal of Clinical Epidemiology. junho de 2019;110:12–22.

The authors make no comment at all in the introduction about the usefulness or pitfalls of the models for death prediction for Chagas disease population already available in the literature. I suggest looking for models predicting death for Chagas disease, as I am aware of at least three of them, and comment about the potential improvement required in the field from this point on. 

Methods:

“Of the 2,161 participants at baseline, 123 202 with negative or indeterminate serology for CD were excluded, thus all participants included in 124 this study (n=1,959) had confirmed serology.” should be in the “study population” section. Often, it is confusing writing reports of studies that are actually sub studies mentioning a main study. Take special attention to mention key points applicable to substudy only or main cohort only. I suggest the authors clearly state which are the methods from the main cohort and the ones applied on this report.

Results:

There is no scale value or axis label for the vertical axis at figure 2. Is it really supposed to be like this or there is something missing?

Discussion:

I believe there is two main results to be discussed from this research. the first one is the applicability of the chosen models which would be when and how these models could be used and alternatively when and how these models could not be used, or the limitations impairing their use. Additionally, how better or worse these models are when comparing to the existing models. This discussion is not present. The second main result is briefly commented in discussion. The authors stated that BNP alone is the strongest predictor. This is not exactly new as for long many researchers have shown that heart function expressed as ejection fraction is the is strongest predictor of death for this population. Additionally, it has a performance to identify death in two ears similar 2 many other combinations in the model and the authors cited a research where BNP was tested as prognostic marker showing similar performance to the observed in this research. So, what are the differences (advantages and disadvantages) of applying BNP as a single test, and applying BNP in the machine learning model? Of course, this discussion only makes sense if there is expectation to use BNP for new patients in the future. If this is not the case, it wouldn't make any sense to test BNP in a prediction model in first place. 

One point that should be in discussion section is what courses of actions are possible from these model results? What decisions health care providers can make or improve from the application of these models? Let's suppose a physician is making a prediction with one of these models and the model returns that this patient will die in the next two years. What could this physician do with this information?

What are the future research to improve the results shown in this report and the field of Chagas disease mortality prevention?

Kind regards and may the force be with you.

Reviewer #2: I do not recommend the publication of the work in the present form. 

While I believe that the data and the study are interesting and should be communicated in some form, I do not believe that the analysis reported in the manuscript is of sufficient rigour. My suggestion to the authors is to reconsider the perspective of the manuscript and present the analysis from the point of view of an exploratory analysis of an interesting cohort, rather than a predictive modelling effort. 

It is also important that the authors realise that there is a place and role for the use of machine learning approaches in the clinical practice. Their use is critical and it should always be supported by extremely sober statistical evidence: Classification errors have a very high cost in the clinical practice, they should not be treated lightly. Failing this evidence, I trust that machine learning is a powerful way to explore the complexity of neglected diseases and I encourage the authors to restrict their use to that task only.

Reviewer #3: The authors present an original, well-structured article with relevant information. Objectives of the study clearly articulated with a clear testable hypothesis stated. Study design appropriate to address the stated objectives. Population clearly described. Sample size sufficient to ensure adequate power to address the hypothesis being tested. There are no concerns about ethical or regulatory requirements.

PLOS authors have the option to publish the peer review history of their article (what does this mean?). If published, this will include your full peer review and any attached files.

Reviewer #1: Yes: Pedro Emmanuel Alvarenga Americano do Brasil

Reviewer #2: No

Reviewer #3: No
---

## [Decision Letter · Decision Letter 1]

25 Mar 2022

Dear Ferreira,

We are pleased to inform you that your manuscript 'Two-year death prediction models among patients with Chagas Disease using machine learning-based methods' has been provisionally accepted for publication in PLOS Neglected Tropical Diseases.

Best regards,

Alberto Novaes Ramos Jr

Associate Editor

Bruce Lee

Deputy Editor

Reviewer's Responses to Questions

**Key Review Criteria Required for Acceptance?**

**Methods**

-Are the objectives of the study clearly articulated with a clear testable hypothesis stated?

-Is the study design appropriate to address the stated objectives?

-Is the population clearly described and appropriate for the hypothesis being tested?

-Is the sample size sufficient to ensure adequate power to address the hypothesis being tested?

-Were correct statistical analysis used to support conclusions?

-Are there concerns about ethical or regulatory requirements being met?

Reviewer #1: (No Response)

Reviewer #2: (No Response)

Reviewer #3: The authors present a well-structured study, with an adequate presentation of results and a detailed statistical analysis.

**Results**

-Does the analysis presented match the analysis plan?

-Are the results clearly and completely presented?

-Are the figures (Tables, Images) of sufficient quality for clarity?

Reviewer #1: (No Response)

Reviewer #2: (No Response)

Reviewer #3: The authors present a well-structured study, with an adequate presentation of results and a very detailed statistical analysis.

**Conclusions**

-Are the conclusions supported by the data presented?

-Are the limitations of analysis clearly described?

-Do the authors discuss how these data can be helpful to advance our understanding of the topic under study?

-Is public health relevance addressed?

Reviewer #1: (No Response)

Reviewer #2: (No Response)

Reviewer #3: Several modifications have been made, with a significant improvement in the quality of the manuscript.

**Editorial and Data Presentation Modifications?**

Reviewer #1: (No Response)

Reviewer #2: (No Response)

Reviewer #3: Several modifications have been made, with a significant improvement in the quality of the manuscript.

**Summary and General Comments**

Reviewer #1: The authors stated that the cohort "had a cardiac abnormality compatible with CD" thus considered elegible. However, the follow-up time was not yet discussed. There are two major points: (1) age may be considered as a proxy of time of disease as one may assume that most if not all subjects were infected in childhood, thus different ages ate the beginning of the follow-up may indicate different opportunities of Chagas disease progression and should be (and already is) adjusted accordingly; (2) It is important to have an idea of the cardiac function severity at the beginning of follow-up (LV ejection fraction), through echocardiography data or risk group data (A B C or D), as it has been proved consistently to be a strong predictor in the past researches. This is important because the more severe cases are likely to progress to death faster. Of course the BNP peptide is an indicator of this phenomena, but it is not as well stablished as LV EF, and there is no discussion regarding this correspondence in this report.

From predictors selection strategy, the authors seemed to choose fewer predictors, 10, from the overall 45 available predictors. ML models usually automatically choose predictors from a full set of predictors by shrinking the weight of a weak predictor toward zero and keeping the strongest predictors no matter how many are they. But the data pilot usually has low degree of control on this process, often called the black box phenomena. Therefore, it is a bit awkward the statement that "In all cases the configuration with 10 variables obtained the best results in terms of GMean and therefore was the chosen configuration". This statement can be true, however it implies that there was a previous predictors selection (10 out of 45) before adjusting the model. If this is true, what was the rationale of choosing a particular set of ten?

"We believe that it is not necessary to show the performance estimates for this set, as this is not a common practice in this type of study and because we believe that this information will not add new knowledge to the discussion." The authors may choose not to estimate the model performance in all sets, however I disagree with this statement. By showing that the performance is similar in validation and test sets the authors may show that performance travels for new predictions, and the likelyhood of model's optimism is low. Additionally, it would not do any harm, from my point of view, to add performances from all sets.

Reviewer #2: (No Response)

Reviewer #3: The authors present a manuscript of relevant interest. The subject falls within the scope of the journal. The paper is well written and contains useful information. Several modifications have been made, with a significant improvement in the quality of the manuscript. The bibliography is pertinent and current. Therefore, I recommend the approval of the manuscript for publication in this version.

PLOS authors have the option to publish the peer review history of their article (what does this mean?). If published, this will include your full peer review and any attached files.

Reviewer #1: **Yes: **Pedro Emmanuel Alvarenga Americano do Brasil

Reviewer #2: No

Reviewer #3: No

---

## [Editor Report · Acceptance letter]

8 Apr 2022

Dear Dr. Ferreira,

We are delighted to inform you that your manuscript, "Two-year death prediction models among patients with Chagas Disease using machine learning-based methods," has been formally accepted for publication in PLOS Neglected Tropical Diseases.

Best regards,

Shaden Kamhawi

co-Editor-in-Chief

Paul Brindley

co-Editor-in-Chief
